# Dynamic Integration of Background Knowledge in Neural NLU Systems

## Abstract

Common-sense or background knowledge is required to understand natural language, but in most neural natural language understanding (NLU) systems, the requisite background knowledge is indirectly acquired from static corpora. We develop a new reading architecture for the dynamic integration of explicit background knowledge in NLU models. A new task-agnostic reading module provides refined word representations to a task-specific NLU architecture by processing background knowledge in the form of free-text statements, together with the task-specific inputs. Strong performance on the tasks of document question answering (DQA) and recognizing textual entailment (RTE) demonstrate the effectiveness and flexibility of our approach. Analysis shows that our models learn to exploit knowledge selectively and in a semantically appropriate way.

## 1 Introduction

Understanding natural language depends crucially on common-sense or background knowledge, for example, knowledge about what concepts are expressed by the words being read (lexical knowledge), and what relations hold between these concepts (relational knowledge). As a simple illustration, if an agent needs to understand that the statement "King Farouk signed his abdication" is entailed by "King Farouk was exiled to France in 1952, after signing his resignation", it must know (among other things) that *abdication* means *resignation of a king*.

In most neural natural language understanding (NLU) systems, the requisite background knowledge is implicitly encoded in the models' parameters. That is, what background knowledge is present has been learned from task supervision and also by pre-training word embeddings (where distributional information reliably reflects certain kinds of useful background knowledge, such as semantic relatedness). However, acquisition of background knowledge from static training corpora is limiting for two reasons. First, we cannot expect that all background knowledge that could be important for solving an NLU task can be extracted from a limited amount of training data. Second, as the world changes, the facts that may influence how a text is understood will likewise change. In short: building suitably large corpora to capture all relevant information, and keeping the corpus and derived models up to date with changes to the world would be impractical.

In this paper, we develop a new architecture for dynamically incorporating external background knowledge in NLU models. Rather than relying only on static knowledge implicitly present in the training data, supplementary knowledge is retrieved from a knowledge base to assist with understanding text inputs. Since NLU systems must necessarily read and understand text inputs, our approach incorporates background knowledge by repurposing this reading machinery—that is, we read the text being understood together with supplementary natural language statements that assert facts (**assertions**) which are relevant to understanding the content (§2).

Our knowledge-augmented NLU systems operate in a series of phases. First, given the text input that the system must understand, which we call the **context**, a set of relevant supporting assertions is retrieved. While learning to retrieve relevant information for solving NLU tasks is an important question (Nogueira & Cho, 2017; Narasimhan et al., 2016, *inter alia*), in this work, we focus on learning how to incorporate retrieved information, and use simple heuristic retrieval methods to identify plausibly relevant background from an external knowledge base. Once the supplementary texts have been retrieved, we use a word embedding refinement strategy that incrementally reads the context and retrieved assertions starting with context-independent word embeddings and building

successively refined embeddings of the words that ultimately reflect both the relevant supporting assertions and the input context (§3). These **contextually refined word embeddings**, which serve as dynamic memory to store newly incorporated knowledge, are used in any task-specific reading architecture. The overall architecture is illustrated in Figure 1. Although we are incorporating a new kind of information into the NLU pipeline, a strength of this approach is that the architecture of the reading module is independent of the final NLU task—the only requirement is that the final architecture use word embeddings.

We carry out experiments on several different datasets on the tasks of document question answering (DQA) and recognizing textual entailment (RTE) evaluating the impact of our proposed solution with both basic task architectures and a sophisticated task architecture for RTE (§4). We find that our embedding refinement strategy is quite effective (§5). On four standard benchmarks, we show that refinement helps—even refining the embeddings just using the context (and no additional background information) can improve performance significantly, and adding background knowledge helps further. Our results are very competitive, setting a new state-of-the-art on the recent TriviaQA benchmarks which is remarkable considering the simplicity of the chosen task-specific architecture. Finally, we provide a detailed analysis of how knowledge is being used by an RTE system (§6), including experiments showing that our system is capable of making appropriate counterfactual inferences when provided with "false knowledge".

## 2 EXTERNAL KNOWLEDGE AS SUPPLEMENTARY TEXT INPUTS

Knowledge resources make information that could potentially be useful for improving NLU available in a variety different formats, such as (subject, predicate, object)-triples, relational databases, and other structured formats. Rather than tailoring our solution to a particular structured representation, we assume that all supplementary information either already exists in natural language statements or can easily be recoded as natural language. In contrast to mapping from unstructured to structured representations, the inverse problem is not terribly difficult. For example, given a triple *(monkey, isA, animal)* we can construct the free-text assertion "a monkey is an animal" using a few simple rules. Finally, the free-text format means that knowledge that exists only in unstructured text form is usable by our system.

A major question that remains to be answered is: given some text that is to be understood, what supplementary knowledge should be incorporated? The retrieval of contextually relevant information from knowledge sources is a complex research topic by itself, and it is likewise crucially dependent on the format of the underlying knowledge base. There are several statistical (Manning et al., 1999) and more recently neural approaches (Mitra & Craswell, 2017) and approaches based on reinforcement learning (Nogueira & Cho, 2017). In this work we make use of a simple heuristic from which we almost exhaustively retrieve all potentially relevant assertions (see §4), and rely on our reading architecture to learn to extract only relevant information.

In the next section, we turn to the question of how to leverage the retrieved supplementary knowledge (encoded as text) in a NLU system.

## 3 REFINING WORD EMBEDDINGS BY READING

In order to incorporate information from retrieved input texts we propose to compute contextually refined word representations prior to processing the NLU task at hand and pass them to the task in the form of word embeddings. Word embeddings thus serve as a form of memory that not only contains general-purpose knowledge (as in typical neural NLU systems) but also contextual information (including retrieved background knowledge). Our incremental refinement process *encodes* input texts followed by *updates* on the word embedding matrix using the encoded input in multiple reading steps. Words are first represented non-contextually (i.e., standard word type embeddings), which can be conceived of as the columns in an embedding matrix $\mathbf{E}^0$. At each progressive reading step $\ell \geq 1$, a new embedding matrix $\mathbf{E}^\ell$ is constructed by refining the embeddings from the previous step $\mathbf{E}^{\ell-1}$ using (user-specified) contextual information $\mathcal{X}^\ell$ for reading step $\ell$, which is a set of natural language sequences (i.e., texts). An illustration of our incremental refinement strategy can be found in Figure 1.

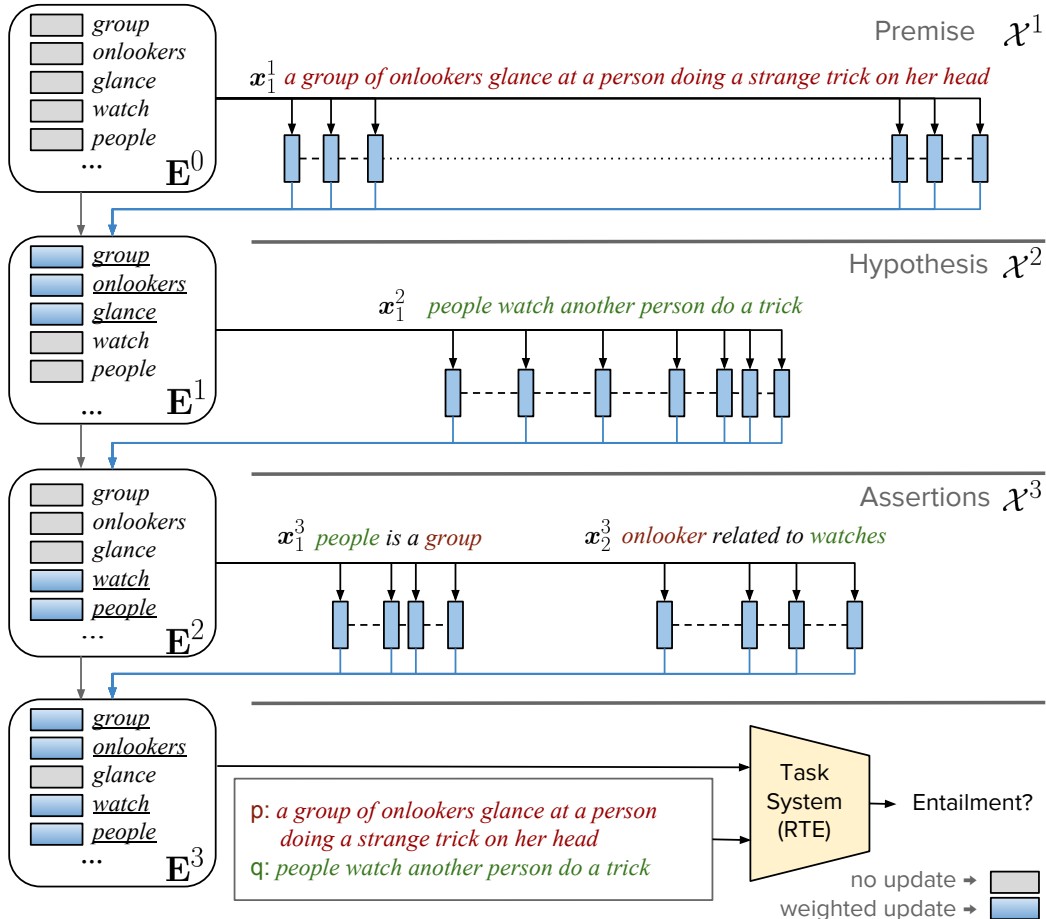

Figure 1: Illustration of our context-dependent, refinement strategy for word representations on an example from the SNLI dataset comprising the premise, hypothesis and additional external information in form of free-text assertions. The reading architecture constructs refinements of word representations incrementally (conceptually represented as columns in a series of embedding matrices) $\mathbf{E}^{\ell}$ are incrementally refined by reading the input text and textual renderings of relevant background knowledge before computing the representations used by the task model (in this figure, RTE).

In the following, we define this procedure formally. We denote the hidden dimensionality of our model by $n$ and a fully-connected layer by $\mathrm{FC}(\mathbf{z}) = \mathbf{W}\mathbf{z} + \mathbf{b}$, $\mathbf{W} \in \mathbb{R}^{n \times m}$, $\mathbf{b} \in \mathbb{R}^n$, $\mathbf{u} \in \mathbb{R}^m$.

## 3.1 UNREFINED WORD EMBEDDINGS ($\mathbf{E}^0$)

The first representation level consists of non-contextual word representations, that is, word representations that do not depend on any input; these can be conceived of as an embedding matrix $\mathbf{E}^0$ whose columns are indexed by words in $\Sigma^*$. The non-contextual word representation $\mathbf{e}^0_w$ for a single word $w$ is computed by using a gated combination of fixed, pre-trained word vectors $\mathbf{e}^p_w \in \mathbb{R}^{n'}$ with learned character-based embeddings $\mathbf{e}^{char}_w \in \mathbb{R}^n$. The formal definition of this combination is given in Eq. 1:

$$\mathbf{e}^{p'}_w = \mathrm{ReLU}(\mathrm{FC}(\mathbf{e}^p_w))$$
$$\mathbf{g}_w = \sigma\left(\mathrm{FC}\begin{bmatrix}\mathbf{e}^{p'}_w \\ \mathbf{e}^{char}_w\end{bmatrix}\right)$$
$$\mathbf{e}^0_w = \mathbf{g}_w \odot \mathbf{e}^{p'}_w + (\mathbf{1} - \mathbf{g}_w) \odot \mathbf{e}^{char}_w. \tag{1}$$

We compute $\mathbf{e}_w^{char}$ using a single-layer convolutional neural network using $n$ convolutional filters of width 5 followed by a max-pooling operation over time. Combining pre-trained with character based word embeddings in such a way is common practice. Our approach follows (Seo et al., 2017; Weissenborn et al., 2017).

## 3.2 Contextually Refined Word Representations ($\mathbf{E}^\ell$, $\ell \geq 1$)

In order to compute contextually refined word embeddings $\mathbf{E}^\ell$ given prior representations $\mathbf{E}^{\ell-1}$ we assume a given set of texts $\mathcal{X}^\ell = \{\boldsymbol{x}_1^\ell, \boldsymbol{x}_2^\ell, \ldots\}$ that are to be read at refinement iteration $\ell$. Each text $\boldsymbol{x}_i^\ell$ is a sequence of word tokens. We embed all tokens of every $\boldsymbol{x}_i^\ell$ using the embedding matrix from the previous layer, $\mathbf{E}^{\ell-1}$. To each word, we concatenate a one-hot vector of length $L$ with position $\ell$ set to 1, indicating which layer is currently being processed.[1] Stacking the vectors into a matrix, we obtain a $\mathbf{X}_i^\ell \in \mathbb{R}^{d \times |\boldsymbol{x}_i^\ell|}$. This matrix is processed by a bidirectional recurrent neural network, a BiLSTM (Hochreiter & Schmidhuber, 1997) in this work. The resulting output is further projected to $\hat{\mathbf{X}}_i^\ell$ by a fully-connected layer followed by a ReLU non-linearity (Eq. 2).

$$\hat{\mathbf{X}}_i^\ell = \text{ReLU}(\text{FC}(\text{BiLSTM}(\mathbf{X}_i^\ell))) \tag{2}$$

To finally update the previous embedding $\mathbf{e}_w^{\ell-1}$ of word $w$, we initially maxpool all representations of occurrences matching the lemma of $w$ in every $\boldsymbol{x} \in \mathcal{X}^\ell$ resulting in $\hat{\mathbf{e}}_w^\ell$ (Eq. 3). Finally, we combine the context-independent representation $\mathbf{e}_w$ with $\hat{\mathbf{e}}_w$ to form a context-sensitive representation $\mathbf{e}_w^\ell$ via a gated addition which lets the model determine how much to revise the embedding with the newly read information (Eq. 5).

$$\hat{\mathbf{e}}_w^\ell = \max \left\{ \hat{\mathbf{x}}_k^\ell \mid \boldsymbol{x}^\ell \in \mathcal{X}^\ell, \text{lemma}(x_k^\ell) = \text{lemma}(w) \right\} \tag{3}$$

$$\mathbf{u}_w^\ell = \sigma \left( \text{FC} \left( \begin{bmatrix} \mathbf{e}_w^{\ell-1} \\ \hat{\mathbf{e}}_w^\ell \end{bmatrix} \right) \right) \tag{4}$$

$$\mathbf{e}_w^\ell = \mathbf{u}_w^\ell \odot \mathbf{e}_w^{\ell-1} + (\mathbf{1} - \mathbf{u}_w^\ell) \odot \hat{\mathbf{e}}_w^\ell \tag{5}$$

Note that we soften the matching condition for $w$ using lemmatization, $\text{lemma}(w)$, during the pooling operation of Eq. 3 because contextual information about certain words is usually independent of the current word form $w$ they appear in. As a consequence, this minor linguistic pre-processing step allows for additional interaction between tokens of the same lemma.

The important difference between our contextual refinement step and conventional multi-layer (RNN) architectures is the pooling operation that is performed over occurrences of tokens that share the same lemma. This effectively connects different positions within and between different texts with each other thereby mitigating the problems arising from long-distance dependencies. More importantly, however, it allows models to make use of additional input such as relevant background knowledge.

## 4 Experimental Setup

We run experiments on four benchmarks for two popular tasks, namely recognizing textual entailment (RTE) and document question answering (DQA). In the following we describe different aspects of our experimental setup in more detail.

**Task-specific Models** Our primary interest is to explore the value of our refinement strategy with relatively generic task architectures. Therefore, we chose basic single-layer bidirectional LSTMs (BiLSTMs) as encoders with a task-specific, feed-forward neural network on top for the final prediction. Such models are common baselines for NLU tasks and can be considered general reading architectures as opposed to the more highly tuned, task-specific NLU systems that are necessary

---

[1] Adding this one-hot feature lets the refinement model to learn update word embeddings differently in different levels.

to achieve state-of-the art results. However, since such models frequently underperform more customized architectures, we also add our refinement module to a reimplementation of a state-of-the-art architecture for RTE called ESIM (Chen et al., 2017b).

All models are trained end-to-end jointly with the refinement module. For the DQA baseline system we add a simple *lemma-in-question* feature (liq) as suggested in Weissenborn et al. (2017) when encoding the context to compare against competitive baseline results. We provide the exact model implementations for our BiLSTM baselines and general training details in Appendix A.

**Question Answering**   We apply our DQA models on 2 recent DQA benchmark datasets, SQuAD (Rajpurkar et al., 2016) and TriviaQA (Joshi et al., 2017). The task is to predict an answer span within a provided document $p$ given a question $q$. Both datasets are large-scale, containing on the order of 100k examples. Because TriviaQA is collected via distant supervision the test set is divided into a large but noisy distant supervision part and a much smaller (on the order of hundreds) human verified part. We report results on both. See Appendix A.1 for implementation details.

**Recognizing Textual Entailment**   We test on the frequently used SNLI dataset (Bowman et al., 2015), a collection of $570k$ sentence pairs, and the more recent MultiNLI dataset ($433k$ sentence pairs) (Williams et al., 2017). Given two sentences, a premise $p$ and a hypothesis $q$, the task is to determine whether $p$ either *entails*, *contradicts* or is *neutral* to $q$. See Appendix A.2 for implementation details.

**Knowledge Source**   We make use of **ConceptNet**[2] (Speer & Havasi, 2012), a freely-available, multi-lingual semantic network that originated from the Open Mind Common Sense project and incorporates selected knowledge from various other knowledge sources, such as Wiktionary[3], Open Multilingual WordNet[4], OpenCyc and DBpedia[5]. It presents information in the form of relational triples.[6]

**Assertion Retrieval**   We would like to obtain information about the relations of words and phrases between $q$ and $p$ from ConceptNet in order to strengthen the connection between the two sequences. Because assertions $a$ in ConceptNet come in form of (subject, predicate, object)-triples $(s, r, o)$, we retrieve all assertions for which $s$ appears in $q$ and $o$ appears in $p$, or vice versa. Because still too many such assertions might be retrieved for an instance, we rank all retrievals based on their respective subject and object. To this end we compute a ranking score which is the inverse product of appearances of the subject and the object in the KB, that is $\mathrm{score}(a) = \left( \sum_{a'} \mathbb{I}(s_{a'} = s_a) \cdot \sum_{a'} \mathbb{I}(o_{a'} = o_a) \right)^{-1}$, where $\mathbb{I}$ denotes the indicator function. This is very related to the popular *idf* score (inverted document frequency) from information retrieval which ranks terms higher that appear less frequently across different documents. During training and evaluation we only retain the top-$k$ assertions which we specify for the individual experiments separately. Note that (although very rarely) it might happen that no assertions are retrieved at all.

**Refinement Order**   When employing our embedding-refinement strategy, we first read the document ($p$) followed by the question ($q$) in case of DQA, and the premise ($p$) followed by the hypothesis ($q$) for RTE, that is, $\mathcal{X}^1 = \{p\}$ and $\mathcal{X}^2 = \{q\}$. Additional knowledge in the form of a set of assertions $\mathcal{A}$ is integrated after reading the task-specific input for both DQA and RTE, that is, $\mathcal{X}^3 = \mathcal{A}$. In preliminary experiments we found that the final performance is not significantly sensitive to the order of presentation so we decided to fix our order as defined above.

| Model | SQuAD Dev | | TriviaQA Wiki Test | | TriviaQA Web Test | |
| --- | --- | --- | --- | --- | --- | --- |
| | F1 | Exact | F1 | Exact | F1 | Exact |
| BiLSTM + liq | 74.3 | 63.7 | 50.9 / 55.1 | 45.2 / 48.8 | 50.9 / 60.0 | 44.2 / 54.7 |
| + reading | 77.8 | 67.3 | 54.1 / 59.6 | 47.5 / 52.2 | 54.7 / 65.9 | 48.1 / 60.3 |
| + reading + knowledge (20) | 79.2 | 69.0 | 54.6 / **60.7** | 48.4 / **55.1** | **56.9** / 67.3 | **50.6** / 61.9 |
| + reading + knowledge (50) | 79.6 | 69.4 | **55.1** / 59.9 | **48.6** / 53.4 | 56.7 / **68.0** | **50.6** / **63.2** |
| SotA Results | 81.8[1] | 73.2[1] | 46.9 / 55.8[3] | 43.2 / 49.3[3] | 48.3 / 57.6[3] | 44.3 / 53.3[3] |
| | **82.8**[2] | **75.6**[2] | 52.9 / 59.5[1] | 46.9 / 54.5[1] | 52.9 / 61.5[1] | 46.7 / 57.0[1] |

Table 1: Results on the SQuAD development set as well as TriviaQA-Wikipedia and -Web test sets for $n = 300-$dimensional models. TriviaQA results are further divided by distant supervision results (left) and human verified results (right). The model using external knowledge is trained with the top-20/50 retrieved ConceptNet assertions. The *liq*-feature (lemma-in-question) is only used for the baseline. [1]Hu et al. (2017), [2]Wang et al. (2017), [3]Pan et al. (2017).

## 5 RESULTS

### 5.1 QUESTION ANSWERING: SQUAD AND TRIVIAQA

Table 1 presents our results on two question answering benchmarks. We report results on the SQuAD development set[7] and the two more challenging TriviaQA test sets which demonstrate that the introduction of our reading architecture helps consistently with additional gains from using background knowledge. Our systems do even outperform current state-of-the-art models on TriviaQA which is surprising given the simplicity of our task-specific architecture and the complexity of the others. For instance, the system of Hu et al. (2017) uses a complex multi-hop attention mechanism to achieve their results. Even our baseline BiLSTM + liq system reaches very competitive results on TriviaQA which is in line with findings of Weissenborn et al. (2017). To verify that it is not the additional computation that gives the performance boosts when using only our reading architecture (without knowledge), we also ran experiments with 2-layer BiLSTMs (+liq) for our baselines which exhibit similar computational complexity to BiLSTM + reading. We found that the second layer even hurts performance. This demonstrates that pooling over word/lemma occurrences in a given context between layers, which constitutes the main difference to conventional stacked RNNs, is a powerful, yet simple technique. In any case, the most important finding of these experiments is that knowledge actually helps considerably with up to 2.2/2.9% improvements on the F1/Exact measures.

### 5.2 RECOGNIZING TEXTUAL ENTAILMENT: SNLI AND MULTINLI

Table 2 shows the results of our RTE experiments. In general, the introduction of our refinement strategy almost always helps, both with and without external knowledge. When providing additional background knowledge from ConceptNet, our BiLSTM based models improve substantially, while the ESIM-based models improve only on the more difficult MultiNLI dataset. Compared to previously published state-of-the-art systems, our models acquit themselves very well on the MultiNLI benchmark, and competitively on the SNLI benchmark. In parallel to this work, Gong et al. (2017) developed a novel task-specific architecture for RTE that achieves slightly better performance on MultiNLI than our ESIM+reading+knowledge based models.[8] It is worth observing that with our knowledge-enhanced embedding architecture, our generic BiLSTM-based task model outperforms ESIM on MultiNLI, which is architecturally much more complex and designed specifically for the RTE task. Finally, we remark that despite careful tuning, our re-implementation of ESIM fails to

---

[2]http://conceptnet.io/

[3]http://wiktionary.org/

[4]http://compling.hss.ntu.edu.sg/omw/

[5]http://dbpedia.org/

[6]We exclude ConceptNet 4 assertions created by only one contributor and from Verbosity to reduce noise.

[7]Due to restrictions on code sharing, we are not able to use the public evaluation server to obtain test set scores for SQuAD. However, for the remaining tasks we report both development accuracy and held-out test set performance.

[8]Our reading+knowledge refinement architecture can be used of course with this new model.

| Model | SNLI | | MNLI Matched | | MNLI Mismatched | |
|---|---|---|---|---|---|---|
| | Dev | Test | Dev | Test | Dev | Test |
| BiLSTM | 84.4 | 83.4 | 70.0 | 69.8 | 70.2 | 69.4 |
| + reading | 86.1 | 85.4 | 75.3 | 75.8 | 76.3 | 74.9 |
| + reading + knowledge | 86.5 | 85.7 | 76.8 | 77.3 | 77.5 | 76.3 |
| ESIM | 88.2 | 87.2 | 76.8 | 76.3 | 77.3 | 75.8 |
| + reading | 88.0 | 87.3 | 77.8 | 77.8 | 78.4 | 77.0 |
| + reading + knowledge | 87.8 | 87.3 | 78.8 | 78.2 | 78.8 | 77.0 |
| SotA Results | − | 88.0[3] | − | 74.5[2] | − | 73.5[2] |
| | − | **88.6**[1] | − | **78.8**[3] | − | **77.8**[3] |

Table 2: Results on the SNLI as well as MultiNLI-Matched and -Mismatched for $n = 300-$dimensional models. The model using external knowledge is trained with the top-20 retrieved ConceptNet assertions. [1]Chen et al. (2017b), [2]Gong et al. (2017). [3]Chen et al. (2017a).

| Dim/Data | 1/1k | 3/3k | 10/10k | 30/30k | 100/100k | 300/Full |
|---|---|---|---|---|---|---|
| | | | SNLI | | | |
| ESIM | 52.2 | 59.3 | 69.7 | 75.5 | 81.3 | 88.2 |
| + reading | 60.9(+8.7) | 66.0(+6.7) | 71.0(+1.3) | 75.0(-0.5) | 80.6(-0.7) | 88.0(-0.2) |
| + reading + knowledge | 62.9(+2.0) | 68.6(+2.6) | 73.8(+2.8) | 77.4(+2.4) | 81.3(+0.7) | 87.8(-0.2) |
| | | | MultiNLI Matched + Mismatched | | | |
| ESIM | 44.3 | 50.0 | 55.5 | 61.9 | 68.1 | 76.9 |
| + reading | 51.8(+7.5) | 55.8(+5.8) | 60.1(+4.6) | 65.0(+3.1) | 70.7(+2.6) | 78.1(+1.2) |
| + reading + knowledge | 52.4(+0.6) | 57.9(+2.1) | 62.4(+2.3) | 66.6(+1.6) | 71.3(+0.6) | 78.8(+0.7) |

Table 3: Development set results when reducing training data and embedding dimsensionality with PCA. In parenthesis we report the relative differences to the respective result directly above.

match the 88% reported in Chen et al. (2017b); however, with MultiNLI, we find that our implementation of ESIM performs considerably better (by approximately 5%). The instability of the results suggests, as well as the failure of a custom RTE-architecture to consistently perform well suggests that current SotA RTE models may be overfit to the SNLI dataset.

## 5.3 Reducing Training Data & Dimensionality of Pre-trained Word Embeddings

We find that there is only little impact when using external knowledge on the RTE task when using a more sophisticated task model such as ESIM. We hypothesize that the attention mechanisms within ESIM jointly with powerful, pre-trained word representations allow for the recovery of some important lexical relations when trained on a large dataset. It follows that by reducing the number of training data and impoverishing pre-trained word representations the impact of using external knowledge should become larger.

To test this hypothesis, we gradually impoverish pre-trained word embeddings by reducing their dimensionality with PCA while reducing the number of training instances at the same time.[9] Our joint data and dimensionality reduction results are presented in Table 3. They show that there is indeed a slightly larger benefit when employing background knowledge in the more impoverished settings with largest improvements over using only the novel reading architecture when using around 10k examples and reduced dimensionality to 10. However, we observe that the biggest overall impact over the baseline ESIM model stems from our contextual refinement strategy (*reading*) which is especially pronounced for the 1k and 3k experiments. This highlights once more the usefulness of our refinement strategy even without the use of additional knowledge.

---

[9]Although reducing either embedding dimensionality or data individually exhibit similar (but less pronounced) results we only report the joint reduction results here.

| **p**: | His off-the-cuff style *seems* amateurish [...] | the *net* cost of operations. | but uh these guys [...] file their uh their *final* exams [...] |
| **h**: | He didn't *look like* an amateur | The *gross* cost. | These men filed their *midterm* exams [...] |
| **a**: → | look like synonym seem contradiction | gross antonym net contradiction | midterm antonym final contradiction |
| **ā**: → | look like antonym seem entailment | gross synonym net entailment | midterm synonym final entailment |

Table 4: Three examples for the *antonym ↔ synonym* swapping experiment on MultiNLI. **p**-premise, **h**-hypothesis, **a**-assertion, **ā**-swapped assertion.

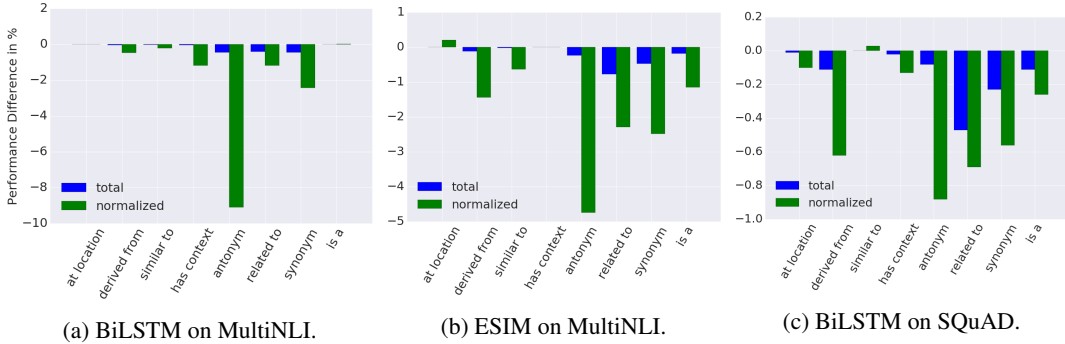

(a) BiLSTM on MultiNLI. (b) ESIM on MultiNLI. (c) BiLSTM on SQuAD.

Figure 2: Performance differences when ignoring certain types of knowledge, i.e., relation predicates during evaluation. Normalized performance differences are measured on the subset of examples for which an assertion of the respective relation predicate occurs.

## 6 ANALYSIS OF KNOWLEDGE UTILIZATION

**Is additional knowledge used?** To verify whether and how our models make use of additional knowledge, we conducted several experiments. First, we evaluated models trained with knowledge on our tasks while not providing any knowledge at test time. This ablation drops performance by 3.7–3.9% accuracy on MultiNLI, and by 4% F1 on SQuAD. This indicates the model is refining the representations using the provided assertions in a useful way.

**Are models sensitive to semantics of the provided knowledge?** The previous result does not show that the models utilize the provided assertions in any consistent way (it may just reflect a mismatch of training and testing conditions). Therefore, to test our models sensitivity towards the semantics of the assertions, we run an experiment in which we swap the *synonym* with the *antonym* predicate in the provided assertions during test time. Because of our heuristic retrieval mechanism, not all such "counterfactuals" will affect the truth of the inference, but we still expect to see a more significant impact. The performance drop on MultiNLI examples for which either a *synonym* or an *antonym*-assertion is retrieved is about **10%** for both the BiLSTM and the ESIM model. This very large drop clearly shows that our models are sensitive to the semantics of the provided knowledge. Examples of prediction changes are presented in Table 4. They demonstrate that the system has learned to trust the presented assertions to the point that it will make appropriate counterfactual inferences—that is, the change in knowledge has *caused* the change in prediction.

**What knowledge is used?** After establishing that our models are somehow sensitive to semantics we wanted to find out which type of knowledge is important for which task. For this analysis we exclude assertions including the most prominent predicates in our knowledge base individually when evaluating our models. The results are presented in Figure 2. They demonstrate that the biggest performance drop in total (blue bars) stems from *related to* assertions. This very prominent predicate appears much more frequently than other assertions and helps connecting related parts of the 2 input sequences with each other. We believe that *related to* assertions offer benefits mainly from a modeling perspective by strongly connecting the input sequences with each other and thus

bridging long-range dependencies similar to attention. Looking at the relative drops obtained by normalizing the performance differences on the actually affected examples (green) we find that our models depend highly on the presence of *antonym* and *synonym* assertions for all tasks as well as partially on *is a* and *derived from* assertions. This is an interesting finding which shows that the sensitivity of our models is selective wrt. the type of knowledge and task. The fact that the largest relative impact stems from *antonyms* is very interesting because it is known that such information is hard to capture with distributional semantics contained in pre-trained word embeddings.

# 7 RELATED WORK

The role of background knowledge in natural language understanding has long been remarked on, especially in the context of classical models of AI (Schank & Abelson, 1977; Minsky, 2000); however, it has only recently begun to play a role in neural network models of NLU (Ahn et al., 2016; Xu et al., 2016; Long et al., 2017; Dhingra et al., 2017). However, previous efforts have focused on specific tasks or certain kinds of knowledge, whereas we take a step towards a more general-purpose solution for the integration of heterogeneous knowledge for NLU systems by providing a simple, general-purpose reading architecture that can read background knowledge encoded in simple natural language statements, e.g., "abdication is a type of resignation".

Bahdanau et al. (2017) use textual word definitions as a source of information about the embeddings of OOV words. In the area of visual question answering Wu et al. (2016) utilize external knowledge in form of DBpedia comments (short abstracts/definitions) to improve the answering ability of a model. Marino et al. (2017) explicitly incorporate knowledge graphs into an image classification model. Xu et al. (2016) created a recall mechanism into a standard LSTM cell that retrieves pieces of external knowledge encoded by a single representation for a conversation model. Concurrently, Dhingra et al. (2017) exploit linguistic knowledge using MAGE-GRUs, an adapation of GRUs to handle graphs, however, external knowledge has to be present in form of triples. The main difference to our approach is that we incorporate external knowledge in free text form on the word level prior to processing the task at hand which constitutes a more flexible setup. Ahn et al. (2016) exploit knowledge base facts about mentioned entities for neural language models. Bahdanau et al. (2017) and Long et al. (2017) create word embeddings on-the-fly by reading word definitions prior to processing the task at hand. Pilehvar et al. (2017) seamlessly incorporate information about word senses into their representations before solving the downstream NLU task, which is similar. We go one step further by seamlessly integrating all kinds of fine-grained assertions about concepts that might be relevant for the task at hand.

Another important aspect of our approach is the notion of dynamically updating word-representations. Tracking and updating concepts, entities or sentences with dynamic memories is a very active research direction (Kumar et al., 2016; Henaff et al., 2017; Ji et al., 2017; Kobayashi et al., 2017). However, those works typically focus on particular tasks whereas our approach is task-agnostic and most importantly allows for the integration of external background knowledge. Other related work includes storing temporary information in weight matrices instead of explicit neural activations (such as word representations) as a biologically more plausible alternative.

# 8 CONCLUSION

We have presented a novel task-agnostic reading architecture that allows for the dynamic integration of background knowledge into neural NLU models. Our solution, which is based on the incremental refinement of word representations by reading supplementary inputs, is flexible and be used with virtually any existing NLU architecture that rely on word embeddings as input. Our results show that embedding refinement using both the system's text inputs, as well as supplementary texts encoding background knowledge can yield large improvements. In particular, we have shown that relatively simple task architectures (e.g., based on simple BiLSTM readers) can become competitive with state-of-the-art, task-specific architectures when augmented with our reading architecture.

## ACKNOWLEDGMENTS

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

# A  IMPLEMENTATION DETAILS

In the following we explain the detailed implementation of our two task-specific, baseline models. We assume to have computed the contextually (un-)refined word representations depending on the setup and embedded our input sequences $\boldsymbol{q} = (q_1, ..., q_{L_Q})$ and $\boldsymbol{p} = (p_1, ..., p_{L_P})$ to $\mathbf{Q} \in \mathbb{R}^{n \times L_Q}$ and $\mathbf{P} \in \mathbb{R}^{n \times L_P}$, respectively. The word representation update gate in Eq. 4 is initialized with a bias of 1 to refine representations only slightly in the beginning of training. In the following as before, we denote the hidden dimensionality of our model by $n$ and a fully-connected layer by $\text{FC}(\mathbf{z}) = \mathbf{Wz} + \mathbf{b}, \mathbf{W} \in \mathbb{R}^{n \times m}, \mathbf{b} \in \mathbb{R}^n, \mathbf{u} \in \mathbb{R}^m$.

## A.1  QUESTION ANSWERING

**Encoding**  In the DQA task $\boldsymbol{q}$ refers to the question and $\boldsymbol{p}$ to the supporting text. At first we process both sequences by identical BiLSTMs in parallel (Eq. 6) followed by separate linear projections (Eq. 7) .

$$\hat{\mathbf{Q}} = \text{BiLSTM}(\mathbf{Q}) \qquad \hat{\mathbf{P}} = \text{BiLSTM}(\mathbf{P}) \qquad \hat{\mathbf{Q}} \in \mathbb{R}^{2n \times L_Q}, \hat{\mathbf{P}} \in \mathbb{R}^{2n \times L_P} \qquad (6)$$

$$\tilde{\mathbf{Q}} = \mathbf{U}_Q \hat{\mathbf{Q}} \qquad \tilde{\mathbf{P}} = \mathbf{U}_P \hat{\mathbf{P}} \qquad \tilde{\mathbf{Q}} \in \mathbb{R}^{n \times L_Q}, \tilde{\mathbf{P}} \in \mathbb{R}^{n \times L_P}, \qquad (7)$$

$\mathbf{U}_Q, \mathbf{U}_P \in \mathbb{R}^{n \times 2n}$ are initialized by $[I; I]$ where $I \in \mathbb{R}^{n \times n}$ is the identity matrix.

**Prediction**  Our prediction– or answer layer is the same as in Weissenborn et al. (2017). We first compute a weighted, $n$-dimensional representation $\tilde{\boldsymbol{q}}$ of the processed question $\tilde{\mathbf{Q}}$ (Eq. 8).

$$\alpha = \text{softmax}(\mathbf{v}_q \tilde{\mathbf{Q}}) \quad , \mathbf{v}_q \in \mathbb{R}^n$$
$$\tilde{\mathbf{q}} = \sum_i \alpha_i \tilde{\mathbf{q}}_i \qquad (8)$$

The probability distribution $p_s/p_e$ for the start/end location of the answer is computed by a 2-layer MLP with a ReLU activated, hidden layer $\boldsymbol{s}_j$ as follows:

$$\mathbf{s}_j = \text{ReLU}\left(\text{FC}_s\left(\begin{bmatrix} \tilde{\mathbf{p}}_j \\ \tilde{\mathbf{q}} \\ \tilde{\mathbf{p}}_j \odot \tilde{\mathbf{q}} \end{bmatrix}\right)\right) \qquad \mathbf{e}_j = \text{ReLU}\left(\text{FC}_e\left(\begin{bmatrix} \tilde{\mathbf{p}}_j \\ \tilde{\mathbf{q}} \\ \tilde{\mathbf{p}}_j \odot \tilde{\mathbf{q}} \end{bmatrix}\right)\right)$$
$$p_s(j) \propto \exp(\mathbf{v}_s \mathbf{s}_j) \qquad \mathbf{v}_s \in \mathbb{R}^n \qquad\qquad p_e(j) \propto \exp(\mathbf{v}_e \mathbf{s}_j) \qquad \mathbf{v}_e \in \mathbb{R}^n \qquad (9)$$

The model is trained to minimize the cross-entropy loss of the predicted start and end positions, respectively. During evaluation we extract the span $(i, k)$ with the best span-score $p_s(i) \cdot p_e(k)$ of maximum token length $k - i = 16$.

## A.2  RECOGNIZING TEXTUAL ENTAILMENT

**Encoding**  Analogous to DQA we encode our input sequences by BiLSTMs, however, for RTE we use conditional encoding (Rocktäschel et al., 2015) instead. Therefore, we initially process the embedded hypothesis $\mathbf{Q}$ by a BiLSTM and use the respective end states of the forward and backward LSTM as initial states for the forward and backward LSTM that processes the embedded premise $\mathbf{P}$.

**Prediction**  We concatenate the outputs of the forward and backward LSTMs processing the premise $\boldsymbol{p}$, i.e., $\left[\tilde{\mathbf{p}}_t^{fw}; \tilde{\mathbf{p}}_t^{bw}\right] \in \mathbb{R}^{2n}$ and run each of the resulting $L_P$ outputs through a fully-connected layer with ReLU activation ($\mathbf{h}_t$) followed by a max-pooling operation over time resulting in a hidden state $\mathbf{h} \in \mathbb{R}^n$. Finally, $\mathbf{h}$ is used to predict the RTE label as follows:

$$\mathbf{h}_t = \text{ReLU}\left(\text{FC}\left(\begin{bmatrix}\tilde{\mathbf{p}}_t^{fw}\\\tilde{\mathbf{p}}_t^{bw}\end{bmatrix}\right)\right)$$

$$\mathbf{h} = \underset{t}{\text{maxpool}}\ \mathbf{h}_t$$

$$p(c) \propto \exp(\mathbf{v}_c\mathbf{h})\quad, \mathbf{v}_c \in \mathbb{R}^n \tag{10}$$

The probability of choosing category $c \in \{\text{entailment, contradiction, neutral}\}$ is defined in Eq. 10. Finally, the model is trained to minimize the cross-entropy loss of the predicted category probability distribution $p$.

## A.3 TRAINING

As pre-processing steps we lowercase all inputs and tokenize it. Additionally, we make use of lemmatization as described §3.2 in which is necessary for matching. As pre-trained word representations we use 300-dimensional word-embeddings from `Glove` (Pennington et al., 2014). We employed ADAM (Kingma & Ba, 2015) for optimization with an initial learning-rate of $10^{-3}$ which was halved whenever the F1 measure (DQA) or the accuracy (RTE) dropped on the development set between 1000/2000 minibatches for DQA and RTE respectively. We used mini-batches of size 16 for DQA and 64 for RTE. Additionally, for regularization we make use of dropout with a rate of 0.2 on the computed non-contextual word representations $\mathbf{e}_w$ defined in §3.1 with the same dropout mask for all words in a batch. All our models were trained with 3 different random seeds and the top performance is reported.

