# OpenReview forum: "Dynamic Integration of Background Knowledge in Neural NLU Systems"
_ICLR.cc/2018/Conference — Reject_

### Official Review · AnonReviewer2 · 2017-11-26
**A model that incorporates background knowledge to improve on multiple NLU tasks**

**Rating:** 5
**Confidence:** 3

**Review:**

This paper proposes a model for adding background knowledge to natural language understanding tasks. The model reads the relevant text and then more assertions gathered from background knowledge before determining the final prediction. The authors show this leads to some improvement on multiple tasks like question answering and natural language inference (they do not obtain state of the art but improve over a base model, which is fine in my opinion).

I think the paper does a fairly good job at doing what it does, it is just hard to get excited by it.
Here are my major comments:

* The authors explains that the motivation for the work is that one cannot really capture all of the knowledge necessary for doing natural language understanding because the knowledge is very dynamic. But then they just concept net to augment text. This is quite a static strategy, I was assuming the authors are going to use some IR method over the web to back up their motivation. As is, I don't really see how this motivation has anything to do with getting things out of a KB. A KB is usually a pretty static entity, and things are added to it at a slow pace.

* The author's main claim is that retrieving background knowledge and adding it when reading text can improve performance a little when doing QA and NLI. Specifically they take text and add common sense knowledge from concept net. The authors do a good job of showing that indeed the knowledge is important to gain this improvement through analysis. However, is this statement enough to cross the acceptance threshold of ICLR? Seems a bit marginal to me.

* The author's propose a specific way of incorporating knowledge into a machine reading algorithm through re-embeddings that have some unique properties of sharing embeddings across lemmas and also having some residual connections that connect embeddings and some processed versions of them. To me it is unclear why we should use this method for incorporating background knowledge and not some simpler way. For example, have another RNN read the assertions and somehow integrate that. The process of re-creating embeddings seems like one choice in a space of many, not the simplest, and not very well motivated. There are no comparisons to other possibilities. As a result, it is very hard for me to say anything about whether this particular architecture is interesting or is it just in general that background knowledge from concept net is useful. As is, I would guess the second is more likely and so I am not convinced the architecture itself is a significant contribution.

So to conclude, the paper is well-written, clear, and has nice results and analysis. The conclusion is that reading background knowledge from concept net boost performance using some architecture. This is nice to know but I think does not cross the acceptance threshold.

---

> ### Author Response · Authors · 2017-12-05
> **RE: Reviewer2**
>
> We thank you for your review and helpful  comments.
>
> == Dynamic Integration ==
> We agree with your point that ConceptNet is a static resource. Despite this fact, we show that our model can deal with changing knowledge in section 6, by swapping antonym with synonym relations. This results in our model making correct counterfactual decisions and shows that our model is not finetuned on a static resource. Extending ConceptNet would have a positive impact on our models without retraining. This is why we call it dynamic.
>
> In this paper we tried to address on the fly integration of knowledge first, because this is a non trivial problem by itself. We agree that (once such a system works) it would be interesting to focus more on the retrieval part (potentially from the web).
>
> == Marginality of Contribution ==
> Showing that explicit, external knowledge can be used to improve reading comprehension can have a huge impact. This means that task models can focus on handling knowledge given to the system from the outside rather than memorizing every single fact about the world in a limited set of parameters. Furthermore, it allows the model to react to changing knowledge dynamically without having to be retrained. We demonstrate this ability in section 6 where our model learns to make correct counterfactual decisions. This is an important first step in that direction.
>
> Given that this is the first work that tries and succeeds in integrating knowledge in such a general setting (i.e., knowledge provided in plain text), we believe that this work opens up very promising paths for future research.
>
> == Simpler way of integrating knowledge ==
> "For example, have another RNN read the assertions and somehow integrate that. "  -> This is exactly what we are doing. However, integrating this information in a simpler way than we did through word embeddings seems unlikely, especially when considering that we aim for a task agnostic reading architecture complementary to any neural task architecture. This goal was also clearly motivated in the paper, namely, that we want to be task and model agnostic with our approach such that we stack any task model on top of our reading module.
>
> Finally, this paper does not claim that our architecture is the best solution, but it is the *first* solution to incorporate knowledge almost seamlessly and we show that it works. That is why we stuck with that architecture and focused on analyzing how a model with knowledge is able to perform and handle the provided knowledge.

---

### Official Review · AnonReviewer1 · 2017-11-26
**Adding knowledge is good idea; this may be too simple a way to do it?**

**Rating:** 5
**Confidence:** 4

**Review:**

The main emphasis of this paper is how to add background knowledge so as to improve the performance of NLU (specifically QA and NLI) systems. They adopt the sensible perspective that background knowledge might most easily be added by providing it in text format. However, in this paper, the way it is added is simply by updating word representations based on this extra text. This seems too simple to really be the right way to add background knowledge.

In practice, the biggest win of this paper turns out to be that you can get quite a lot of value by sharing contextualized word representations between all words with the same lemma (done by linguistic preprocessing; the paper never says exactly how, not even if you read the supplementary material). This seems a useful observation which it would be easy to apply everywhere and which shows fairly large utility from a bit of linguistically sensitive matching!  As the paper notes, this type of sharing is the main delta in this paper from simply using a standard deep LSTM (which the paper claims to not work on these data sets, though I'm not quite sure couldn't be made to work with more tuning).

pp. 6-7: The main thing of note seems to be that sharing of representations between words with the same lemma (which the tables refer to as "reading" is worth a lot (3.5-6.0%), in every case rather more than use of background knowledge (typically 0.3-1.5%). A note on the QA results: The QA results are certainly good enough to be in the range of "good systems", but none of the results really push the SOTA. The best SQuAD (devset) results are shown as several percent below the SOTA. In the table the TriviaQA results are shown as beating the SOTA, and that's fair wrt published work at the time of submission, but other submissions show that all of these results are below what you get by running the DrQA (Chen et al. 2017) system off-the-shelf on TriviaQA, so the real picture is perhaps similar to SQuAD, especially since DrQA is itself now considerably below the SOTA on SQUAD. Similar remarks perhaps apply to the NLI results.

p.7 In the additional NLI results, it is interesting and valuable to note that the lemmatization and knowledge help much more when amounts of data (and the covarying dimensionality of the word vectors) is much smaller, but the fact that the ideas of this paper have quite little (or even negative) effects when run on the full data with full word vectors on top of the ESIM model again draws into question whether enough value is being achieved from the world knowledge.

Biggest question:
 - Are word embeddings powerful enough as a form of memory to store the kind of relational facts that you are accessing as background knowledge?

Minor notes:
 - The paper was very well written/edited. The only real copyediting I noticed was in the conclusion: and be used ➔ and can be used; that rely on ➔ that relies on.
 - Should reference to (Manning et al. 1999) better be to (Manning et al. 2008) since the context here appears to be IR systems?
 - On p.3 above sec 3.1: What is u? Was that meant to be z?
 - On p.8, I'm a bit suspicious of the "Is additional knowledge used?" experiment which trains with knowledge and then tests without knowledge. It's not surprising that this mismatch might hurt performance, even if the knowledge provided no incremental value over what could be gained from standard word vectors alone.
 - In the supplementary material the paper notes that the numbers are from the best result from 3 runs. This seems to me a little less good experimental practice than reporting an average of k runs, preferably for k a bit bigger than 3.

---

> ### Author Response · Authors · 2017-12-05
> **RE: Reviewer1**
>
> We thank you for your review and helpful  comments.
>
> We will clarify that we used spacy for lemmatization. Sorry for missing that.
> We will also clarify misunderstandings in the paper as well as change the reference.
>
>
> ==Are word embeddings powerful enough as a form of memory to store the kind of relational facts that you are accessing as background knowledge?==
>
> Short answer: it might be and frankly, we show that it works (see section 6 for counterfactual reasoning analysis).
>
> To understand the reasons for this one has to consider that only facts that are relevant for the context and task have to be remembered on the fly for each concept. These are not too many, and their relevance can change from task to task.
>
> ==Simplicity==
> 'This seems too simple to really be the right way to add background knowledge. '
> We disagree with this statement, because 1) our solution works and 2) what does 'right' mean? Obviously, there are infinite solutions to this problem and we present merely one not claiming that it is the 'right' way. However, it is a very *practical* and *useful*  way of doing it because it is orthogonal to all existing task-architectures operating on word embeddings (almost all). Furthermore, we focus on simplicity to have more control over the system, because it is the first that integrates background knowledge in textual form like this. Because we are the first to try this, we like to point out that this is not trivial, although the solution might seem that way (which is not a bad sign).
>
> We agree, however, that there might be other interesting architectural options for integrating knowledge which should be explored in the future.
>
> ==Reading as biggest win==
> We acknowledge that this is the case and another indicator that our reading architecture is a good solution to incorporate contextual information from the task and back knowledge alike. However, this does not change the fact that background knowledge helps, although less than merely reading. Please note that the reason for this is that 1) additional knowledge is not always present and 2) knowledge helps resolving long-tail phenomena such as handling antonyms properly.
>
> ==Results==
> Our QA results are not SotA and we do not aim for that since this requires a lot of engineering and trial and error to get 1 or 2% more. On the NLI datasets we show that also more complex models such as ESIM can improve when employing our strategy and using knowledge.
> Comparing against SotA is of course important but it never gives the full picture because every setup is slightly different and small changes can lead to rather large performance differences (from our experience with these datasets).
>
> = SQuAD =
> Extra knowledge gives a boost of an additional 2% which is quite remarkable given that this merely stems from the use of external knowledge (note that external knowledge can mostly help to account for long tail phenomena).
> In particular, considering that we are using a single Layer BiLSTM as task model, no attention, no hand crafted features, nothing,  the reported results are remarkably high.
>
> In contrast, DrQA explicitly hand engineers features (NER, word in question, etc.) using sophisticated pre-trained systems such as NER systems and uses a three layer BiLSTM. Still, our model achieves higher or comparable performance.
>
> =TriviaQA=
> TriviaQA is somewhat delicate and reported results are not entirely comparable to each other, because data preparation is different. However, we will adopt recent advances in [1] to  deal properly with TriviaQA and update results accordingly.
>
> [1] Clark et al. Simple and Effective Multi-Paragraph Reading Comprehension
>
> =NLI results=
> We present slightly negative results for SNLI. We argued in the paper that systems are overfit on SNLI and that SNLI is very homogeneous wrt the vocabulary used while being very large. This means that most knowledge can be learnt implicitly on that amount of data.  However, this is not the case anymore when lowering the data as shown in section 5.3.
>
> In short, SNLI is not a good test bed for our work, however, we included results for reasons of transparency.
>
> On MultiNLI we respectfully disagree with the reviewer by noting that knowledge helps in most cases, up to several % when considering simpler BiLSTM as task model.
>
> == Is additional knowledge used? ==
> This is a valid experiment because in many cases in NLI there is NO knowledge retrieved by our heuristic, that means our model knows how to deal with the absence of it. However, at the same time it learns to rely on it, hence the drop in performance which shows that knowledge is indeed utilized in some meaningful way.
>
> == best result from 3 runs==
> Results did only change slightly between runs (up to 0.2%), so there would not have been a big difference in presentation. We agree, however, that this might indeed be good practice, but frankly, common practice on these datasets is unfortunately to only report the best results.

---

> ### Author Response · Authors · 2017-12-05
> **"Too simple"**
>
> Thanks for the thorough review. I would make two points:
> 1) Although both reading and knowledge add to this task (and reading adds more in some cases), both are important, new results.
> 2) NLP/NLU/ML/(vision/AI/...) are full of models that are "too simple". We should be very glad about this. First, it means we can solve some interesting, potentially complex tasks without terribly sophisticated models. Second, they serve as a reasonable baseline for significant architectural innovations.

---

### Official Review · AnonReviewer3 · 2017-11-27
**This paper proposes a new method to generate vector representations of text and background knowledge. The propsed method is evaluated on QA and textual entailment (TE) tasks.**

**Rating:** 6
**Confidence:** 4

**Review:**

The quality of this paper is good. The presentation is clear but I find lack of description of a key topic. The proposed model is not very innovative but works fine for the DQA task. For the TE task, the proposed method does not perform better than the state-of-the-art systems.

- As ESIM is one of the key components in the experiments, you should briefly introduce ESIM and explain how you incorporated with your vector representations into ESIM.
- The reference of ESIM is not correct.
- Figure 1 is hard to understand. What do you indicate with the box and arrow? Arrows seem to have some different meanings.
- What corpus did you use to pre-train word vectors?
- As the proposed method was successful for the QA task, you need to explain QA data sets and how the questions are solved.
- I also expect performance and  error analysis of the task results.
- To claim "task-agnostic", you need to try to apply your method to other NLP tasks as well.
- Page 3. \Sigma is not defined.

---

> ### Author Response · Authors · 2017-12-05
> **RE: Reviewer3**
>
> We thank you for your review and helpful  comments.
>
> We are sorry that parts of our paper were not clear enough and will improve upon that given your concerns.
>
> == Clarification==
>
> The paper is not about "generating vector representations of text and background knowledge", it is about incorporating explicit knowledge (that is text) into a neural NLU system when given a single instance of a task, such that it is able to make better predictions. Knowledge is provided from the outside and the system learns to make use of it. We do this by refining word representations on-the-fly given additional knowledge in textual form when processing a single instance of a task. That means, we do not generate vector representations for knowledge nor text, but simply learn to process and use it on-the-fly.
>
> == ESIM ==
> We will check and update the reference. As mentioned in our paper, we build refined word emebeddings. Because ESIM takes word emebeddings as input naturally (as most neural NLU systems) there is no need to adapt ESIM at all. Being agnostic to the task architecture is an important aspect of our solution.
>
> == Pre trained Embeddigns ==
> We do not pre train any word embeddings. We use pretrained embeddings from Glove (Pennington et al 2015) and refine these on the fly using our architecture.
>
> ==  Analysis==
> we provided several analysis with respect to our contribution, namely integrating background knowledge. Note that this work does not aim at improving SotA on highly competitive datasets but tries to explore a way to incorporate background knowledge on the fly. This is why we did not focus on task-specifics in our analysis.
>
> == agnostic ==
> our solution is task and model-agnostic. We show that it works well together with 3 different models on 4 different datasets of 2 challenging NLU tasks.

---

### Decision · Program_Chairs · 2018-01-29
**ICLR 2018 Conference Acceptance Decision**

**Decision:**

Reject

**Comment:**

Pros:
+ The paper is very clearly written.
+ The proposed re-embedding approach is easily implemented and can be integrated into fancier architectures.

Cons:
- A lot of the gains reported come from lemmatization, and the gains from background knowledge become marginal when used on a stronger baseline (e.g., ESIM with full training data and full word vectors).

This paper is rather close to the decision boundary. The authors had reasonable answers for some of the reviewers' concerns, but in the end the reviewers were not completely convinced.